# OpenReview forum: "Can In-Context Reinforcement Learning Recover From Reward Poisoning Attacks?"
_ICLR.cc/2026/Conference — Submitted to ICLR 2026_

### Official Review · Reviewer_baJq · 2025-10-29

**Soundness:** 3
**Presentation:** 4
**Contribution:** 3
**Rating:** 4
**Confidence:** 4

**Summary:**

The paper addresses adversarial robustness from reward poisoning attacks for in-context reinforcement learning, focusing on the decision pretrained transformer (DPT) framework. The paper introduces an adversarial training framework to find optimally worst-case perturbations for the adversary to make to the target's rewards, and find that training DPT against this adversary makes the model robust.

**Strengths:**

The paper develops a strong formal baseline for adversarial robustness in in-context reinforcement learning, a growing and increasingly pertinent field of study. They demonstrate the effectiveness of adversarial training in the newer setting, expanding upon an important conclusion found in older works [1,2].

The paper is well written, clearly describes implementation details, and motivates design choices.




[1] Aleksander Madry, Aleksandar Makelov, Ludwig Schmidt, Dimitris Tsipras, Adrian Vladu: Towards Deep Learning Models Resistant to Adversarial Attacks. ICLR (Poster) 2018

[2] Anay Pattanaik, Zhenyi Tang, Shuijing Liu, Gautham Bommannan, Girish Chowdhary: Robust Deep Reinforcement Learning with Adversarial Attacks. AAMAS 2018: 2040-2042

**Weaknesses:**

#### Motivation
- Given the success of adversarial training in past work, its direct application to DPT seems straightforward, and its effectiveness unsurprising. Further, the comparisons to prior work mention a difference in setting (in-context, multi-task) but do not relate these differences to a motivation. Thus, it is not immediately clear how AT-DPT addresses challenges present in prior works.

#### Evaluation
- Tables 1 & 2 state that lower regret is better; however report a higher regret for the proposed methods in clean and random environments as compared to baselines. This is noted in the text, but it is not discussed as to why this might happen. For clean settings, this usually amounts to a form of catastrophic forgetting. The disparity under random attacks is counterintuitive and should be discussed.
- The proposed method maximizes reward as an optimization criterion; however, the experiments mostly discuss regret when compared to other robust baselines. It is worth noting the raw performance of the proposed method versus other robust methods.
- Table 4 shows raw performance, but excludes robust baselines.

**Questions:**

- Is there an intuitive explanation (or a good guess) as to why AT-DPT underperforms against random attacks specifically?
- The proposed method uses adversaries that perturb the reward model. Can it be applied directly to a setting with observation-perturbing adversaries?

#### Main Question
- How does AT-DPT address challenges in prior works?
The score may be revised upwards if this information is provided.

---

> ### Author Response · Authors · 2025-11-24
>
> We greatly appreciate your time spent reviewing our work and providing detailed and valuable feedback.
>
> ## Motivation
> In general, robustness to test-time poisoning attacks is important for agents relying on in-context learning capabilities. An example that also considers the multi-task setting is LLM-based agents, whose context can be easily poisoned if they use external tools or RAG mechanisms, as demonstrated by prior work. Our work is generally important for understanding how to enable robustness to poisoning attacks for agents that exhibit in-context RL capabilities -- as shown by [1], this includes LLMs.
>
> ## Evaluation
> We thank the reviewer for identifying this discrepancy. Table 1 contains an error -- it reports results from an earlier version of the paper. The correct results appear in the Appendix, Table 5 (page 18), or,  see Table 1 in the new revision of the paper.
> The corrected data shows AT-DPT achieves 38.7 regret on uniform random attacks (not 45.4), similar to DPT with 37.2. This shows that random attacks affect every algorithm similarly, and adversarial training does not degrade performance on them.
> We would like the reviewer to consider that in Table 4 the robust baseline is NPG. The Robust Policy Gradient paper ([1], Section 4) shows NPG is robust to $\varepsilon$-contamination when the rewards are not arbitrarily large. In our case we experiment with bounded corruption, and also observe the robustness of NPG in experiments. We refer to this claim in our paper (line 449), although further discussion could make it more clear. The result obtained from NPG indeed displays this robustness, confirming the theory.
>
> ## Key Challenges
> From the provided references, we provide the challenges distinct in our work:
> - Pattanaik et al. (2018) shows adversarial training working well within a single task environment. We extend this to a multi-task setting (i.e., both solve different tasks _and_ be robust to reward poisoning) -- AT-DPT learns a single policy for multiple tasks.
> - Pattanaik et al. (2018) shows an attack which only affects the immediate next state. Whereas we consider an attack that poisons the context and may influence future actions. We argue this is a harder problem, because attacks can compound across multiple timesteps.
> - Mądry et al. (2018) states that if an attack is robust to PGD adversaries, it would be robust against many different types of adversaries. In our case we show this principle extends to in-context learning -- the first row in Table 1 shows that AT-DPT can generalize to multiple different attacks while only being trained on one type.
>
> To motivate our contributions and novelty better, please see the response to Reviewer 4dN9 (Contributions & Novelty).
>
> In addition, to make our contribution stronger we provide some theoretical insights in a general response to all reviewers.
>
> ## Observation Perturbing Adversaries
> We believe our setting could indeed be extended to adversaries which perturb observations. In that setup we would have to define a budget for the observation, and similarly use an RL algorithm for the attacker to minimize the episode reward. We note that attacks on observations may face other issues, such as temporal consistency. This may be easily solved with information in the context, e.g., for a poisoned observation at timestep t the victim may simply ‘interpolate’ between t-1 and t+1 as a naive verification, and the attack might perform poorly.
>
> ## Minor Issues
> Regret is commonly discussed in bandit literature, and we adopt this metric for DPT in the bandit setting as well.
>
> We kindly ask the reviewer to consider this response. Thank you again for your feedback and please let us know if you have any additional comments or questions.
>
>
> [1] Monea et al. (2025) LLMs Are In-Context Bandit Reinforcement Learners. CoLM 2025

---

> > ### Comment · Reviewer_baJq · 2025-11-26
> >
> > Thanks to the authors for the clarifying response. The concerns about the motivation and evaluation sections are resolved.
> >
> > Regarding the key challenges in prior works, there is still some uncertainty regarding precisely how AT-DPT specifically addresses these challenges. Specifically, the multi-task aspect: given that LLMs are generally seen as multi-task learners [3], and that adversarial attacks can be viewed as tasks, as an adversarial policy can affect whole trajectories [4], it seems that improvements to robustness in this case would be a result of the LLM's ability to learn new tasks in-context.
> > 1) Does AT-DPT contain any mechanism targeting multi-task learning beyond the inherent LLM ability?
> >
> > Similarly: while the initial review references [1,2] as seminal baselines, the paper would benefit from clear distinctions between AT-DPT and closely related recent works. Specifically:
> >
> > 2) Algorithm Distillation and extensions (Laskin 2022, Zisman 2024) are mentioned as relevant ICRL prior works, but are not present in the experiments. Is there a reason they are excluded?
> >
> > 3) The primary concern is that the insight that _supplying DPT with adversarial examples increases robustness_ is trivial in isolation. Is there empirical evidence that some aspect of AT-DPT (e.g., the Nash $(\theta^\*, \phi^\*)$) that induces robustness more than a naive extension to DPT, say, fine-tuning against random perturbations?
> >
> >
> >
> > [1] Madry et. al., _Towards Deep Learning Models Resistant to Adversarial Attacks._ (ICLR 2018)
> >
> > [2] Pattanaik et. al., _Robust Deep Reinforcement Learning with Adversarial Attacks._ (AAMAS 2018)
> >
> > [3] Radford et.al., _Language Models are Unsupervised Multitask Learners._ (OpenAI)
> >
> > [4] Gleave et. al., _Adversarial Policies: Attacking Deep Reinforcement Learning._ (ICLR 2020)

---

> > > ### Author Response · Authors · 2025-12-03
> > >
> > > Thank you for your comment! We appreciate the acknowledgement that some of your concerns have been resolved. Please find our response to your additional questions below.
> > >
> > > 1. Although we consider a GPT-2 style transformer architecture, we emphasize that we do _not_ rely on large-language-model (LLM) capabilities in our setting. In particular, we do not attempt to elicit in-context learning abilities from an LLM. Instead, following prior work on Decision Transformers [1], we pretrain a decision transformer using data collected from a given set of tasks. This model can be interpreted as a history-dependent policy $\pi_{\theta}(\cdot|s, h)$ where $h$ denotes the history of interactions (including rewards). This history dependence allows the model to implement a learning algorithm and function as a meta-learner. However, as shown in our results on DPT, this meta-learner, while robust to random noise in $h$, is **not** robust to adversarial noise (in our case, corrupted rewards). Intuitively, our approach introduces worst-case adversarial perturbations in $h$ (via reward manipulation) in order to robustify the resulting meta-learner. Because in our setting, such adversarial corruptions are both task-dependent and policy-dependent (the latter arising from the absence of assumptions such as convex–concave structure), we train a **population** of adversaries (one per task) together with the policy $\pi_{\theta}$. \
> > > \
> > > We reiterate the distinction between our setting and the reviewer’s suggestion. Although investigating corruption-robust in-context RL capabilities of LLMs would indeed be interesting, to our knowledge such capabilities have not been extensively studied (see [2] and references therein). Consequently, it is not currently clear whether existing LLMs can be considered in-context reinforcement learners that are robust to adversarial noise.
> > >
> > > 2. Regarding Algorithm Distillation (AD), this is a good observation! We don’t implement it as a baseline, as AD is designed to imitate a learning (source) algorithm (see Section 3.1 in [3]). Hence, we don’t expect it to perform better than the source algorithm, which in our case would be corruption-robust baselines whose performance we already report in the results. In contrast, DPT tries to predict an optimal action, so it may potentially lead to **emergent strategies** at test-time, as argued in [1]:
> > > > While DPT also leverages autoregressive SL, it does not distill an existing RL algorithm in order to imitate how to learn. Instead, we pretrain DPT to predict optimal actions, yielding potentially emergent online and offline strategies at test time that automatically leverage the task structure to behave similarly to posterior sampling.
> > >
> > > 3. Thank you for the suggestion. As we already explained above, DPT is somewhat robust to noise, but not adversarial noise. We have managed to train the AT-DPT against a random adversary (due to time constraints only for the bandit setting), and present our results in Table 1 (2nd row -- _AT-DPT (r)_) in the new revision of the paper. These results show that training against random noise improves robustness in many cases, but is substantially worse than training against a strong adversary. Hence, for better robustness it is important to train a strong adversary.
> > > These results are obtained from the same setup as the other results in the table -- training for 20 rounds, fraction of steps poisoned $\varepsilon = 0.4$; and we remind that we evaluate each algorithm on the attackers it has not seen during training.
> > >
> > >
> > > Thank you again for your comment and for engaging in this discussion. We hope we have addressed your concern about key challenges.
> > >
> > > [1] Lee et al., Supervised Pretraining Can Learn In-Context Reinforcement Learning, NeurIPS 2023
> > >
> > > [2] Monea et al. (2025) LLMs Are In-Context Bandit Reinforcement Learners. CoLM 2025
> > >
> > > [3] Laskin et al, In-context Reinforcement Learning with Algorithm Distillation, ICLR 2023

---

### Official Review · Reviewer_LCz1 · 2025-10-30

**Soundness:** 3
**Presentation:** 3
**Contribution:** 3
**Rating:** 4
**Confidence:** 4

**Summary:**

This paper addresses reward poisoning attacks. In-context learning has been particularly popular and in specific Few-shot learning has achieved great accolades in this field. Just by observing a handful of data it can adapt to multiple tasks of that type. However, adversarial attacks and reward poisoning affect the policy output and this paper proposes an efficient algorithm to recover from such reward poisoning attacks.

**Strengths:**

1. The technique is very nice, and the motivation is very clear
2. The extensions of the algorithm to a wide variety of settings depict its effectiveness.

**Weaknesses:**

1. Page 3, under section 3.1, the description of the reward function might not have a distribution over real numbers. I get it that the paper aligns the reward to a Normal or Gaussian distribution, but in general, making it a probability simplex is not very justified and might contrast with normal terminology. Same goes for $\pi^{\dagger}_{\phi}$ in page 4

2. Algorithm 1 line 4. It might be M and not $\mathcal{M}$

3. It might be helpful to know why is it necessary for DPT parameters to be frozen

4. Why is the experiments restricted to only one environment? The extension to other environments such as Machine replacement, River swim, Frozen Lake in tabular settings, etc. or Cartpole, Mountain car, etc., 1 or 2 other environments would give a major understanding. Moreover there are much more possibilities for introducing randomness there.

**Questions:**

1. It might be interesting to see how AT-DPT performs against robust variants of RL algorithms like Robust variants of Q-learning or Robust Natural Actor Critic variants, upon viewing the effectiveness in the Bandit setting.

2. What happens when the architecture of the attacker and the Agent is exactly the same?
3. What happens when extended to other environments?

---

> ### Author Response · Authors · 2025-11-24
>
> Thank you for the review and the set of constructive comments and questions. We are glad to see that you find our method nice, the motivation clear, and that we demonstrated the effectiveness of our algorithm by testing it in a variety of settings.
>
> Q1. It seems that there is no available implementation of Robust Q-learning. We did find an implementation for Robust Natural Actor Critic [1], and if time permits would hope to include it in the comparison.
>
> Thank you for the suggestion. It seems that there is no available implementation of Robust Q-learning. For Natural Actor Critic, robust variants do exist. Our current MDP evaluation includes NPG, which has theoretical robustness guarantees (Zhang et al. 2021), and AT-DPT performs comparably (Table 4). We believe the current evaluation provides sufficient evidence for AT-DPT's robustness, although we acknowledge that additional robust baselines would further strengthen the empirical case. If time and space permit in the final version, we would be interested in expanding this comparison.
>
> Q2. We consider the adaptive attacker’s architecture to be the same as the victim architecture (line 395), and the performance we observe shows AT-DPT is performing well (Table 2).
>
> Q3. In addition to Darkroom2 ($5 \times 5$ grid), we run additional experiments for the MDP setting on the Miniworld environment (3D navigation; also considered by the original DPT work, Lee et al. 2023), see Table 9 in the Appendix. We chose these to match the original DPT paper for fair comparison. In Miniworld the result is similar to Darkroom2 -- with a corruption ratio $\varepsilon = 0.2$ or greater AT-DPT performs better than the frozen DPT baseline and PPO.
>
> In addition to the answers above, we provide some theoretical insights to support our paper in a general response to all reviewers.
>
> ## Minor Issues
> Reward distribution notation: $R: S \times A \to \Delta(\mathcal{R})$ is standard in stochastic RL (e.g., Sutton & Barto). The reward is a random variable, therefore defined as a distribution.
> Why is DPT frozen -- We compare the effect of adversarial training. "DPT ❄" uses only pre-training (no adversarial training), showing that corruption robustness comes from our training procedure, not properties inherent to DPT.
>
> We would be grateful if you could let us know if your concerns are addressed, and if there are still outstanding issues or questions you have with the paper.
>
> [1] Zhou et al. (2023). Natural actor-critic for robust reinforcement learning with function approximation. NeurIPS 2023

---

### Official Review · Reviewer_4dN9 · 2025-10-31

**Soundness:** 3
**Presentation:** 2
**Contribution:** 3
**Rating:** 4
**Confidence:** 2

**Summary:**

This paper focuses on test-time reward poisoning attacks against in-context reinforcement learning, where an adversary manipulates the reward signals observed by the agent during inference to alter its in-context learning behavior. To mitigate this issue, this paper adapts adversarial training to the in-context RL setting, proposing the Adversarially Trained Decision-Pretrained Transformer (AT-DPT).

This framework can be formalized as a bi-level optimization problem. At the inner level, an adversarial attacker is optimized to learn a reward-corruption policy that perturbs the rewards within a soft budget so as to degrade the agent’s true return and expose its vulnerability. At the outer level, the transformer-based agent (DPT) is optimized to recover the correct actions and maintain high performance despite the corrupted feedback observed in its context. By alternating these two objectives, AT-DPT effectively learns to perform in-context adaptation that is resilient to both non-adaptive and adaptive reward-poisoning strategies.

Experiments across bandit and MDP environments demonstrate that this adversarially trained model substantially improves robustness compared to standard and corruption-robust RL baselines.

**Strengths:**

1. The paper tackles an interesting question concerning the robustness of in-context reinforcement learning under test-time reward poisoning. This threat model has received little attention so far, and the authors provide a clear and well-defined solution to address it.

2. The proposed AT-DPT framework extends adversarial training to the in-context setting. The idea is clear and makes sense. The bi-level setup, where the attacker changes the rewards and the agent learns to handle these changes, matches well with how in-context learning works.

3. The empirical evaluation is thorough, covering several environments and both adaptive and non-adaptive adversaries. Among these experiments, the results consistently show that AT-DPT outperforms both standard and robust baselines, indicating that the proposed method effectively enhances robustness.

**Weaknesses:**

1. While the paper introduces a new problem setting, the core idea of applying adversarial training to improve robustness is not particularly new. Extending this idea to the in-context reinforcement learning scenario is interesting, but the paper does not clearly explain what makes this adaptation non-trivial. As a result, it is difficult to determine whether the paper makes a genuinely non-trivial contribution or merely presents a straightforward adaptation.

2. The paper mainly shows that AT-DPT works empirically, but it lacks a deeper analysis of why it works. There is no theoretical explanation or detailed experimental study that helps us understand the source of its robustness. A more in-depth analysis, either theoretical or empirical, would make the contribution much stronger and more convincing.

3. The paper does not discuss enough about the real-world relevance of the proposed problem. It would be helpful to provide concrete examples or scenarios showing where test-time reward poisoning could actually happen. Without such discussion, it is hard to judge how realistic or important this threat model is in practice.

**Questions:**

1. Could the authors further clarify what specific challenges make the adaptation of adversarial training to the in-context reinforcement learning setting non-trivial?

2. Could the authors provide a deeper analysis to justify why AT-DPT works?

---

> ### Author Response · Authors · 2025-11-24
>
> Thank you for the response! We are happy to see that the reviewer appreciates our research question and framework, and considers experimental evaluation to be thorough.
>
> ## Contributions & Novelty
> Our work contributes not just an algorithm for robustly training decision pre-trained transformers, but also the framework itself. In fact, the framework we introduce is the key contribution of the work. We would like to clarify the following:
>
> a) **Contributions to Adversarial RL**: Prior work on reward poisoning attacks primarily focuses on training-time attacks. However, our work focuses on meta-RL, i.e., learning a learning algorithm. In our setting, poisoning attacks occur at _test-time_, not training-time. Hence, the goal of this work is to provide a principled approach to designing learning algorithms robust to corruptions. As we argue below (in response to your comments about relevance), this can enable us to design more efficient algorithms that account for the distribution of tasks and utilize this information as a prior to obtain better performance. Yet, poisoning attacks in RL have primarily been considered standard RL settings. Note that our approach significantly outperforms baselines that don't account for the multi-task structure (in some cases, e.g., Table 3, by more than 90%!). We believe that there are two reasons for this. First, the algorithm can make use of an informed prior over the distribution of tasks, making corruption schemes that are implausible according to the prior ineffective. Second, the algorithm can be thought of as posterior sampling that uses 'corrected' posterior, which accounts for the corruption process. See our discussion about the theoretical insights for more details. These insights are non-trivial and we consider them as an important contribution to the line of work on adversarial RL.
>
> b) **Contributions to In-Context RL**: We are not aware of works prior to ours that considered poisoning attacks for in-context RL. Hence, our framework is novel relative to this line of work as well. While we agree that our method extends DPT by considering adversarial training, we don't fully understand the reviewer why this would be a weakness of the work.
> In fact, we would like to argue that this is a principled way of designing robust DPT. AT-DPT itself is not a trivial adaptation of DPT. Compared to DPT, it additionally utilizes online RL with similar to population-based training in the adversarial training part of the algorithm. Note that adversarial training is non-trivial in this context, especially for the case of non-adaptive attackers: we train 200 different attacker models simultaneously. Moreover, compared to DPT, which in most cases only manages to approximately match the performance of relevant baselines (in bandit environments), we in many cases substantially improve upon them, making the finding from our experimental results non-trivial.
>
> ## Relevance
> In general, robustness to test-time poisoning attacks is important for agents relying on in-context learning capabilities. A prime example are LLM-based agents, whose context can be easily poisoned if they use external tools or RAG mechanisms, as demonstrated by prior work. Our work is generally important for understanding how to enable robustness to poisoning attacks for agents that exhibit in-context RL capabilities -- as shown by [1], this includes LLMs.
>
> Now, as we explained in one of our earlier comments, one of our goals was to provide a principled approach to learning an RL algorithm robust to corruption which better utilises prior knowledge. Namely, we can tailor this RL algorithm to specific tasks of interest (if our prior is informed), which may yield significant performance gains. Our empirical results support this intuition (e.g., see Table 3).
>
> ## Theoretical Insights
> We provide theoretical insights in a general response, which should strengthen the contribution. We kindly ask the reviewer to consider that response.
>
> We would be grateful if you could let us know if we have addressed your main questions and clarified the important points.
>
>
> [1] Monea et al. (2025) LLMs Are In-Context Bandit Reinforcement Learners. CoLM 2025

---

### Official Review · Reviewer_HT3j · 2025-11-02

**Soundness:** 2
**Presentation:** 2
**Contribution:** 1
**Rating:** 2
**Confidence:** 2

**Summary:**

The paper studies adversarial in-context RL, where the reward signal can be corrupted by a possibly adaptive attacker. It proposes an attacker that aims to minimize the expected return on a task, and a robust variant of the Decision Pretrained Transformer (AT-DPT) to maintain performance against such attacks across different target tasks, which is validated empirically.

**Strengths:**

1. The work formalizes a reward poisoning model for in-context learners and introduces a simple min–max training procedure to achieve robustness against both nonadaptive and adaptive attackers.

**Weaknesses:**

1. The paper focuses on empirical corruption robustness of AT-DPT without theoretical guarantees. This is unusual given the well-specified corruption model. At minimum, for the bandit setting emphasized in the experiments, it would be important to prove that regret degrades at most linearly in the corruption budget (in the spirit of guarantees in [1,2,3]). Without such bounds, the contribution feels incomplete.

2. Many baselines are not designed to be adversarially robust, so their underperformance under attacks is expected. In the MDP setting, no robust baseline is included. I recommend adding robust baselines for both bandits and MDPs, and reporting results with appropriately tuned corruption parameters to ensure a fair comparison.

[1] Nika, A., Singla, A. &amp; Radanovic, G.. (2023). Online Defense Strategies for Reinforcement Learning Against Adaptive Reward Poisoning. Proceedings of The 26th International Conference on Artificial Intelligence and Statistics.

[2] Ye, C., Xiong, W., Gu, Q., & Zhang, T. (2023). Corruption-robust algorithms with uncertainty weighting for nonlinear contextual bandits and markov decision processes. In International Conference on Machine Learning (pp. 39834-39863).

[3] Liu, H., Tajdini, A., Wagenmaker, A., & Wei, C. Y. (2024). Corruption-robust linear bandits: Minimax optimality and gap-dependent misspecification. Advances in Neural Information Processing Systems, 37, 24277-24325.

**Questions:**

1. What is the cumulative regret of DPT (and other baselines) as a function of the corruption ratio $\varepsilon$? At what threshold would AT-DPT start outperforming it?
2. AT-DPT is trained against the attacker class defined in Section 3.2. How robust is it to different reward-poisoning mechanisms?

---

> ### Author Response · Authors · 2025-11-24
>
> We greatly appreciate your time spent reviewing our work and providing detailed and valuable feedback.
>
> ## Theoretical Guarantees
> We provide a more detailed discussion of theoretical insights in a general response and would appreciate the reviewer's consideration.
>
> ## Robust Baselines
> We respectfully disagree with the claim that there are no adversarially robust baselines. For the bandit setting we include:
> - RTS (Xu et al. 2024) -- TS designed for adversarial reward poisoning (line 293),
> - crUCB (Niss & Tewari, 2020) -- a UCB style algorithm robust to $\varepsilon$-contamination (line 298),
> - CRLinUCB (Ding et al., 2022) -- robust linear contextual bandits (line 305),
> with comparison of these methods with AT-DPT in Tables 1, 2, and 3.
> For the MDP setting we also compare to NPG, which is also a robust baseline. We kindly ask the reviewer to take a look at the Robust Policy Gradient paper ([1], Section 4), which shows NPG is robust to $\varepsilon$-contamination when the rewards are not arbitrarily large. In our case we experiment with bounded corruption, and also observe the robustness of NPG in experiments. We refer to this claim in our paper (line 449), although further discussion could make it more clear. Table 4 shows AT-DPT performs comparably to NPG, which is a strong validation.
>
> Regarding inclusion of more baselines, we note that the literature usually discusses theoretical algorithms which do not have practical implementations nor empirical results. We agree that it would indeed be interesting to include more of these for comparison, especially given their principled nature, but we believe it is outside the scope of our paper to implement and tune them. We would, however, be happy to include another baseline given a suitable implementation is provided.
>
> Q1. Further experiments showing different values of the corruption ratio $\varepsilon$ can be seen in the Appendix, Tables 5-9. In all settings AT-DPT outperforms the baselines for all $\varepsilon = 0.1,\, 0.2,\, 0.4$ tested, except for Q-learning and PPO in the MDP setting, with $\varepsilon = 0.1$. With $\varepsilon = 0.2$ and above Q-learning and PPO seem to perform worse than AT-DPT.
>
> Q2. In our experiments we show cross-algorithm robustness, meaning that we evaluate e.g., AT-DPT trained against its own attacker on attackers trained for different algorithms; and adaptive-to-non-adaptive robustness; and the robustness across different corruption levels. We acknowledge that our attacks share the $\varepsilon$-contamination model, but we would like to emphasize that we consider the _worst-case_ attacker under a soft-budget constraint, which we expect to be robust to a wide range of attack strategies. Please also see our comments about theoretical insights. That said, unconstrained attack strategies (e.g., targeted reward poisoning attacks with unbounded corruptions [2]), may require different defense mechanisms (e.g., see [3]). We did not consider these in our experiments because their setting is orthogonal to ours.
>
> We would be grateful if you could let us know if there are still questions or outstanding issues you have with the paper.
>
> [1] Zhang et al. (2021) Robust Policy Gradient against Strong Data Corruption. ICML 2021
>
> [2] Rakhsha et al. (2020) Policy teaching via environment poisoning: Training-time adversarial attacks against reinforcement learning. ICML 2020
>
> [3] Banihashem et al. (2023) Defense Against Reward Poisoning Attacks in Reinforcement Learning. TMLR 2023

---

### Author Response · Authors · 2025-11-24
**Theoretical Insights**

We appreciate the suggestion. Our current work indeed focuses on the empirical side of robust in-context learning, but we agree that it would be interesting to theoretically analyze our approach. While providing general theory for the AT-DPT approach is challenging, we can provide some insights based on the theoretical results of Lee et al. [1]. At a high level, our approach can be viewed as solving the following bi-level objective:

$$
\min_{\theta} E_{M_i, D \sim M_i(\pi_{\theta}, \pi_{\phi, i}),s_{\mathrm{query}}} [ \ell(\pi_{\theta}(\cdot | s_{\mathrm{query}}, D ), a^\*) ] \quad \mathrm{s.t.} \quad \pi_{\phi, i} \in \mathrm{BR}(M_i, \pi_{\theta}),
$$
where BR denotes the set of best responses of the adversary given the agent’s policy. Because the AT-DPT adversaries are trained to minimize the agent’s return under a soft budget constraint, the resulting policy $\pi_{\theta}$​ is expected to perform robustly against a range of adversarial strategies. Empirically, we observe that AT-DPT’s performance does not substantially degrade when evaluated against unseen adversaries.

Now, let’s consider what happens an equilibrium $(\theta^\*, \phi^\*)$ of this optimization problem. We have

$$
\theta^* \in \arg \min_{\theta} E_{M_i, D \sim M_i(\pi_{\theta}, \pi_{\phi^*, i}), s_{\mathrm{query}}} \left[ \ell (\pi_{\theta}( \cdot | s_{\mathrm{query}}, D ), a^{\*}) \right].
$$

Hence, Theorem 1 in [1] suggests that $\pi_{\theta^{\*}}$​ implements in-context posterior sampling, but with a _corrected_ posterior that accounts for adversarial data corruption (i.e., $\pi_{\phi^\*}$). As long as the latent reward means remain identifiable under the corruption process, this would suggest that standard posterior sampling analyses may be applicable. In this case, regret would depend on the extent of corruption since increased corruption that reduces reward separability leads to higher regret (by analogy with the Lai & Robbins lower bound [2]). Given that $\pi_{\phi^\*}$ aims to minimize the agent’s return (under a budget constraint), we expect that a similar regret bound would hold when the test-time corruption differs from $\pi_{\phi^\*}$ but satisfies the budget constraint.

The full theoretical treatment would require us to prove that AT-DPT returns an approximate equilibrium policy. We believe that such an analysis is challenging and out of the scope of this paper.


[1] Lee et al. (2023) Supervised Pretraining Can Learn In-Context Reinforcement Learning. NeurIPS 2023.

[2] Lai and Robbins (1985) Asymptotically efficient adaptive allocation rules. Adv. Appl. Math.

---

### Author Response · Authors · 2025-12-03

We thank the reviewers for their valuable comments and suggestions. We are happy to see that they find the paper well-written, our research question interesting, the motivation clear, the proposed technique nice, and the evaluation thorough. We believe we have addressed all the reviews and would like to provide a short summary of the discussion here:
- We provided theoretical insights and reasoned about the success of our method.
- We added further experimental results showing how strong adversaries trained by AT-DPT improves the performance compared to the random attacks.
- We reaffirmed AT-DPT’s strong performance against the robust baselines for both bandit and MDP settings.
- We elaborated on our contributions from the adversarial RL and in-context RL perspectives. We reiterated the novelty of our problem setting and proposed solution. We also discussed the real-world relevance of our problem setting.
- We discussed the rationale for our focus on DPT in contrast to the other in-context RL methods.

We thank the reviewers for their constructive feedback. We hope that additional results, insights and clarifications will help the AC in their decision.

---

### Meta-Review · Area_Chair_Fp3B · 2026-01-08

**Summary:**

This paper studies the robustness of in-context RL against reward poisoning attacks, proposing Adversarially Trained Decision-Pretrained Transformer (AT-DPT). This framework utilizes a min-max adversarial training that simultaneously trains an attacker to minimize the agent's performance and a DPT model to identify optimal policies from corrupted data. Empirical evaluations across various environments, including multi-armed bandits and Markov Decision Processes, demonstrate that AT-DPT significantly improves resilience and outperforms existing robust baselines.

While the authors have clarified that they consider test-time attack, the algorithmic framework is actually not that different from the case of training-time attack.

**Reviewer Concerns:**

- Lack of theoretical guarantees -- partially addressed:  The authors clarified that a full theoretical treatment was outside the scope, but they argued that AT-DPT implements a form of "in-context posterior sampling" with a corrected posterior that accounts for corruption.
- Novelty: Reviewers questioned whether applying adversarial training to DPT was non-trivial, as adversarial training is a well-established technique. The authors clarified that the setting is different from prior work: they focus on meta-RL, and poisoning attacks occur at test-time, not training-time. As pointed out by Reviewer 4dN9, the paper did not gives insight why the adaptation is non-trivial.  While the authors made some conjectures in the rebuttal, they are not verified through deeper ablation studies.  Overall, while the paper has potential, it still lacks some deeper insight about the difference between training-time and test-time attack.

**Reviewer Scores:**

The reviewer scores are 4442. There is no evidence that the reviewers would change their scores given full discussion.

---

### Decision · Program_Chairs · 2026-01-26

Reject